# Polyphenol Analysis via LC-MS-ESI and Potent Antioxidant, Anti-Inflammatory, and Antimicrobial Activities of *Jatropha multifida* L. Extracts Used in Benin Pharmacopoeia

**DOI:** 10.3390/life13091898

**Published:** 2023-09-12

**Authors:** Durand Dah-Nouvlessounon, Michaelle Chokki, Essé A. Agossou, Jean-Baptiste Houédanou, Martial Nounagnon, Haziz Sina, Romana Vulturar, Simona Codruta Heghes, Angela Cozma, Jacques François Mavoungou, Adriana Fodor, Farid Baba-Moussa, Ramona Suharoschi, Lamine Baba-Moussa

**Affiliations:** 1Laboratory of Biology and Molecular Typing in Microbiology, Department of Biochemistry and Cell Biology, Faculty of Sciences and Techniques, University of Abomey-Calavi, Cotonou 05BP1604, Benin; dahdurand@gmail.com (D.D.-N.); houedanou91@gmail.com (J.-B.H.); esperancefirmin@yahoo.fr (M.N.); sina.haziz@gmail.com (H.S.); 2Faculty of Food Science and Technology, University of Agricultural Sciences and Veterinary Medicine Cluj-Napoca, 3-5 Calea Manastur Street, 400372 Cluj-Napoca, Romania; michaellechokki@gmail.com; 3Laboratoire de Microbiologie et de Technologie Alimentaire, FAST, Université d’Abomey-Calavi, 01BP: 526 ISBA-Champ de Foire, Cotonou 01BP188, Benin; fbmouss@yahoo.fr; 4Laboratory of Pharmacology and Improved Traditional Medicines, FAST, Department of Animal Physiology, University of Abomey-Calavi, Cotonou 01BP526, Benin; elvireagossou@gmail.com; 5Department of Molecular Sciences, “luliu Hatieganu” University of Medicine and Pharmacy, 8 Victor Babes, 400012 Cluj-Napoca, Romania; romanavulturar@yahoo.co.uk; 6Department of Drug Analysis, “luliu Hatieganu” University of Medicine and Pharmacy, 6 Louis Pasteur Street, 400012 Cluj-Napoca, Romania; cmaier@umfcluj.ro; 7Internal Medicine Department, 4th Medical Clinic “luliu Hatieganu” University of Medicine and Pharmacy, 400012 Cluj-Napoca, Romania; angelacozma@yahoo.com; 8Department of Microbiology, International University of Libreville, ESSASSA-Libreville Campus, Essassa BP 20411, Gabon; mavoungoujacques@yahoo.fr; 9Clinical Center of Diabetes, Nutrition and Metabolic Diseases, “luliu Hatieganu” University of Medicine and Pharmacy, 400012 Cluj-Napoca, Romania; adifodor@yahoo.com

**Keywords:** plant extract, LC-MS-ESI, bioactive compound, biological activities, Benin

## Abstract

*Jatropha multifida* L., a plant from the *Euphorbiaceae* family, is commonly used in Benin’s traditional medicine due to its therapeutic benefits. This study aims to explore the medicinal efficacy of *Jatropha multifida* L. by evaluating its various biological activities. An initial phytochemical analysis was conducted, following which the polyphenols and flavonoids were quantified and identified using LC-MS-ESI. The antimicrobial efficacy of the extracts was tested using agar diffusion. Their antioxidant capacity was assessed using several methods: DPPH radical reduction, ABTS radical cation reduction, ferric ion (FRAP) reduction, and lipid peroxidation (LPO). Anti-inflammatory activity was determined based on the inhibition of protein (specifically albumin) denaturation. The study identified several phenolic and flavonoid compounds, including 2-Hydroxybenzoic acid, o-Coumaroylquinic acid, Apigenin-apiosyl-glucoside, and luteolin-galactoside. Notably, the extracts of *J. multifida* demonstrated bactericidal effects against a range of pathogens, with Concentration Minimally Bactericidal (CMB) values ranging from 22.67 mg/mL (for organisms such as *S. aureus* and *C. albicans*) to 47.61 mg/mL (for *E. coli*). Among the extracts, the ethanolic variant displayed the most potent DPPH radical scavenging activity, with an IC50 value of 0.72 ± 0.03 mg/mL. In contrast, the methanolic extract was superior in ferric ion reduction, registering 46.23 ± 1.10 µgEAA/g. Interestingly, the water-ethanolic extract surpassed others in the ABTS reduction method with a score of 0.49 ± 0.11 mol ET/g and also showcased the highest albumin denaturation inhibition rate of 97.31 ± 0.35% at a concentration of 1000 µg/mL. In conclusion, the extracts of *Jatropha multifida* L. are enriched with bioactive compounds that exhibit significant biological activities, underscoring their therapeutic potential.

## 1. Introduction

The escalating issue of antibiotic resistance, especially in foodborne pathogens, presents a significant global health challenge [1]. Foods and their production processes can act as reservoirs and transmission routes for antibiotic-resistant bacteria and their corresponding resistance genes, directly impacting public health. It is estimated that antibiotic-resistant bacterial infections result in approximately 700,000 deaths annually worldwide [2].

Besides infectious diseases, inflammation and oxidative stress are implicated in a variety of ailments, encompassing cardiovascular disease, chronic obstructive pulmonary disease, arthritis, neurodegenerative disorders, cancer, diabetes, and several others [3]. The majority of anti-inflammatory agents currently in use are non-steroidal, and these are not devoid of detrimental side effects. The efficacy of non-steroidal anti-inflammatory drugs (NSAIDs) in treating conditions such as cancer is still under debate. Additionally, NSAIDs have been linked to severe complications, including myocardial infarction, gastrointestinal bleeding, and renal failure [4].

Given the increasing microbial resistance to antibiotics and the rise of metabolic diseases linked to oxidative stress and inflammation, there is a growing interest in exploring medicinal plants as alternatives, due to their diverse repertoire of natural compounds. While plants utilize primary metabolites for their growth and development, they also synthesize secondary metabolites, multifunctional compounds chiefly associated with defense mechanisms and environmental interactions [5]. These secondary metabolites, derived from various biosynthetic pathways, can be grouped into key molecular families such as phenolics, terpenes, steroids, alkaloids, and flavonoids. They are not only essential for plant defense against biotic and abiotic stressors but also play significant roles in modulating plant microbiomes [6]. Additionally, plant secondary metabolites have been recognized for their broad spectrum of biological activities, including antibacterial, antifungal, antioxidant, and anti-inflammatory properties.

Benin Republic’s diverse forest and agro-ecosystems host over 162 forest plant species that have found utility in domestic consumption and commerce, and also hold socio-cultural and religious significance [7]. Many of these plants have been traditionally employed in local medicinal practices. One such noteworthy plant is *Jatropha multifida*, a member of the *Euphorbiaceae* family. Found prevalently in various regions such as Asia and Africa [8], references have reported its ancestral use in traditional medicine in several West African countries including Nigeria and Togo, where the population uses the leaves to treat oral thrush, constipation, and fever [9,10]. Other authors [11,12,13] have reported the use of latex for the treatment of gastrointestinal parasitic diseases in humans and animals (anthemintic), serious wounds related to their microbial infection, as well as skin inflammation. Furthermore, the cytotoxic and antiproliferative effects on tumor cells of compounds extracted from *J. multifida* stem have been reported by Das et al. [14].

Popularly known in Benin as “wouèkè”, *J. multifida* is lauded for its multitude of therapeutic properties. This study endeavors to elucidate the therapeutic potential of this plant, sourced from Benin, by assessing some of its biological activities.

## 2. Materials and Methods

### 2.1. Chemical and Microorganism Culture Medium

The extraction solvents used, as well as specific chemical reagents, included PBS, Na_2_CO_3_ ≥ 99.5% (Cas No.497-19-8), AlCl_3_, Folin–Ciocalteu (Cat. No 109001), ferric chloride (Cas No.7705-08-0), potassium acetate, quercetin (Cas No.117-39-5), gallic acid (Cas No. 149-91-7), vanillin (Cas No. 121-33-5), sulfuric acid, hydrochloric acid (HCl), sodium phosphate, ammonium molybdate, and DPPH (2,2-diphenyl-1-picrylhydrazyl, Cas No. 1898-66-4), (Sigma-Aldrich, St. Louis, MO, USA). HPLC-grade solvents were procured from Merck (Darmstadt, Germany). Water purification was carried out using the Direct-Q UV system from Millipore, (St. Louis, MO, USA). High-purity standard compounds were used for HPLC, specifically, gallic acid, chlorogenic acid (with 99% HPLC purity), and rutin (with 99% HPLC purity) (Sigma, St. Louis, MO, USA). Various culture mediums were used, including nutrient broth, Baird-Parker agar, TBX agar, XLD agar, Palcam agar, and Muller Hinton agar (Oxoid Ltd., Basingstoke, Hampshire, UK). All chemicals and reagents used in this study met analytical grade standards.

### 2.2. Plant Material

The leaves of *J. multifida* were harvested from Aglogbè (06°28′58.6″ N, 002°40′41.6″ E), situated in the Oueme department of Benin. A voucher specimen, labeled AAC81108/HNB, was archived at the National Herbarium of Benin, based at the University of Abomey-Calavi in Cotonou, Benin. Following collection, the leaves were air dried at a consistent temperature of 23 ± 2 °C for a duration of 14 days. Afterward, they were ground to a fine powder using a Retsch grinder, model SM 2000/1430/Upm/Smf, from Haan, Germany.

### 2.3. Preliminary Phytochemical Profiling

The phytochemical analysis of *J. multifida* leaf powder was conducted to identify the primary categories of secondary metabolites, including nitrogenous compounds, polyphenolic and terpenic compounds, and glycosides. The methodology employed was based on the protocol previously outlined by Houghton and Raman [15].

### 2.4. Preparation of Plants Extracts

Extracts from *J. multifida* leaves were prepared using five solvents: ethanol, methanol, a water–ethanol mixture (30:70 *v*/*v*), acetone, and dichloromethane. Specifically, 1 g of the powdered leaf material underwent ultrasonication at 35 Hz with 100 mL of each solvent at ambient temperature for a duration of 2 h. The filtrates from each mixture were then collected using Whatman No. 1 filter paper. Subsequently, they were concentrated using a Heidolph Laborota 400 Rotovap rotary evaporator, Schwabach, Germany, and oven dried at 40 °C.

### 2.5. Total Polyphenol Content Determination

The total polyphenol content of *J. multifida* leaf extracts was determined using the Folin–Ciocalteu method [16]. Within a 96-well microplate, a mixture was created using 25 µL of the Folin–Ciocalteu reagent and 10 µL of each respective extract. After an incubation period of 5 min, 25 µL of a 20% sodium carbonate solution was added to the mixture. This was followed by the addition of 140 µL of ultrapure water. Concurrently, a blank was prepared using the same protocol, but the Folin’s reagent was substituted with ultrapure water. A gallic acid solution, ranging from 500 to 0.97 µg/mL, served as a reference, and results are expressed as µg equivalents of gallic acid per mg of the sample. After an incubation of 30 min, absorbance was measured at 760 nm using a multiwell plate reader (Tecan Pro 200, Tecan Trading AG, Männedorf, Switzerland).

### 2.6. Total Flavonoid Content Determination

The total flavonoid content of *J. multifida* leaf extracts was quantified using the aluminum chloride method as previously described [17]. In a 96-well microplate, a reaction mixture was prepared by combining 100 µL of a 2% aluminum chloride solution with 100 µL of each leaf extract. Following a 15-min incubation, the absorbance of the samples was measured at 415 nm using the Tecan Pro 200 multiwell plate reader. Quercetin, with concentrations ranging from 40 to 0.078 µg/mL, served as the reference standard. Results were expressed in terms of µg equivalents of quercetin per mg of the sample.

### 2.7. Condensed Tannin Content Determination

The condensed tannin content in *J. multifida* leaf extracts was determined using the method outlined by Belem-Kabré et al. [18]. In summary, a reaction mixture was created by combining 1 mL of each plant extract (at a concentration of 5 mg/mL) with 2 mL of 1% vanillin dissolved in 70% sulfuric acid. This mixture was then incubated in a water bath set at 20 °C for 15 min. Absorbance readings for the samples were subsequently taken at 500 nm using the Biomate™ 3 Series Spectrophotometers from Thermo Scientific, Germany. The condensed tannin content, represented as T (%), was computed using the provided formula:$$T\left(\%\right)=5.2\times 10^{-2}\times \frac{\left(A\times V\right)}{P}$$
where 5.2 × 10^−2^ = constant in equivalence of cyanidin, A = absorbance, V = extract volume and P = extract weight. Condensed tannin content in the samples was measured in triplicate and expressed as milligrams of cyanidin equivalent (CEq) per gram of extract.

### 2.8. Hydrolyzable Tannin Content Determination

The hydrolyzable tannin content of *J. multifida* leaf extracts was determined using the method described by Belem-Kabré et al. [18]. A reaction mixture was prepared by combining 1 mL of each extract (at a concentration of 5 mg/mL) with 3.5 mL of a reagent consisting of 10^−2^ M ferric chloride (FeCl_3_) in 10^−3^ M hydrochloric acid (HCl). After incubating for 30 s, the absorbance of the samples was measured at 660 nm using the Biomate™ 3 Series Spectrophotometers from Thermo Scientific, Bremen, Germany. The tannin content was then calculated using the following formula:$$T(\%)=\frac{\left(A\times PM\times V\times FD\right)}{Ɛ\;mole\times P}$$
where: A = absorbance, PM = weight of gallic acid (170.12 g/mol), V = volume of extract, FD = dilution factor, ε mole = 2169 (constant in equivalence of gallic acid), P = extract weight.

### 2.9. LC-DAD-ESI-MS Analysis

In this study, the phenolic compound analysis of *J. multifida* leaf extracts was conducted using a sophisticated HPLC-DAD-ESI-MS system. This system was outfitted with a quaternary pump, an autosampler, a DAD detector, and an MS-6110 single-quadrupole API-electrospray detector. Phenolic separation occurred on the Eclipse XDB C18 column (4.6 × 150 mm, 5 µm particle size) from Agilent Technologies, USA, held at a consistent 25 °C. We used a binary gradient comprised of 0.1% acetic acid/acetonitrile (99:1) in distilled water (designated as solvent A) and 0.1% acetic acid in acetonitrile (solvent B). This operated at a flow rate of 0.5 mL/min, adhering to the elution process.

For the MS fragmentation, we employed the ESI (+) mode, scanning a range from 100 to 1200 *m*/*z*. Set parameters included a capillary voltage of 3000 V, a working temperature of 350 °C, and a nitrogen flow rate of 8 L/min. The DAD continuously observed the eluent, recording absorbance spectra between 200 and 600 nm throughout each session. Data analysis was facilitated by the Agilent ChemStation Software (Rev B.04.02 SP1, Palo Alto, Santa Clara, CA, USA).

The identification process for the phenolic compounds involved correlating retention times, UV-visible readings, and mass spectra with four predetermined standards. Flavanol compounds were referenced to a catechin calibration curve (10–200 µg/mL) and results are presented as catechin equivalents (r^2^ = 0.9985). Hydroxycinnamic acids referenced a chlorogenic acid curve, from 10 to 50 µg/mL, expressed as chlorogenic acid equivalents (r^2^ = 0.9937). For flavonols, a quercetin curve (10–200 µg/mL) was utilized, with results in quercetin equivalents (r^2^ = 0.9951). Anthocyanins referenced a cyanidin curve, spanning 10–100 µg/mL, with results conveyed as cyanidin equivalents (r^2^ = 0.9951).

### 2.10. Antimicrobial Activity of J. multifida Leaf Extracts

#### 2.10.1. Microorganisms Strains and Growth Conditions

A total of twelve microbial strains were employed for the sensitivity evaluations. Those extracts displaying activity were subsequently chosen for MIC and CMB assessments. Among these twelve, five were reference strains: two Gram-positive bacteria (*Staphylococcus aureus* ATCC 6538P and *Listeria monocytogenes* ATCC 19114), two Gram-negative bacteria (*Escherichia coli* ATCC 25922 and *Salmonella enteritidis* ATCC 13076), and one yeast (*Candida albicans* ATCC 10231). In addition to these, seven *Staphylococcus* strains, previously isolated from pork by Attein et al. [19], were utilized. These strains, originating from meat, are preserved in the collection of the Laboratory of Biology and Molecular Typing in Microbiology at the University of Abomey-Calavi, Benin.

To refresh these strains, microbial inoculation was conducted in sterile nutrient broth: 10 mL for bacteria, incubated at 37 °C, and for the *C. albicans* yeast, incubated at 30 °C. This was maintained for 24 h. A subsequent microscopic examination ensured the purity of the inoculum. Each strain was then transferred to a respective selective agar medium, poured into sterile Petri dishes. After a 24-h incubation at 37 °C, the morphology of the resultant cultures was confirmed microscopically. From these dishes, several colonies were moved to a sterile saline solution and then adjusted to match McFarland’s turbidity standard (1.5 × 108 CFU/mL) [20]. The distinct agar media deployed included Baird-Parker agar, enriched with Egg Yolk Tellurite Emulsion for *Staphylococcus*; TBX agar for *E. coli*; XLD agar for *Salmonella enteritidis*; and Palcam agar for *Listeria monocytogenes*.

#### 2.10.2. Antibiogram

The disc diffusion method [21] was employed exclusively with *Staphylococcus* strains isolated from meat. For each strain, Petri dishes laden with Muller Hinton Agar were swabbed with 1 mL of the respective inoculum. Excess liquid was absorbed using sterile blotting paper. Subsequently, 2 to 3 Whatman paper discs, each infused with 25 µL of *J. multifida* leaf extracts at a concentration of 20 mg/mL, were placed on the agar. Following incubation at 37 °C over 24 h, the presence of inhibitory zones around the discs was assessed. This procedure was replicated three times for each sample.

#### 2.10.3. Determination of the Minimum Inhibitory Concentration (MIC)

We employed the microdilution method, using resazurin as an indicator of cell viability. Initially, all wells received 100 µL of nutrient broth for bacterial strains and TSB for yeast. Starting with the first wells, a serial two-fold dilution was achieved using 100 μL from each sample down the row. Each diluted sample was then inoculated with 10 µL of the inoculum, standardized to 1.5 × 10^4^ CFU/mL, corresponding to each specific strain. Following this, we incubated the samples for roughly 24 h at 37 °C for bacterial strains and 30 °C for *Candida albicans*. Gentamicin (concentration: 0.04 mg/mL in saline) and fluconazole (concentration: 10 mg/mL) served as control agents. After the incubation, 20 µL of resazurin was introduced to each well. A further incubation ensued under identical conditions for an additional 2 h. The Minimum Inhibitory Concentration (MIC) was identified as the lowest sample concentration where the blue color remained unchanged without turning pink.

#### 2.10.4. Determination of the Minimum Bactericidal/Fungicidal Concentration (MBC/MFC)

The agar medium inoculation method was used to determined the MBC and MFC. In each row, 10 μL of each dilution from the MIC to the highest concentrations of *J. multifida* leaf extracts was inoculated on Mueller–Hinton agar for the bacteria and YPD agar for the yeast. Petri dishes were incubated for 24 h at 37 °C for MBC and 30 °C for MFC. The lowest concentration that prevented the growth of bacteria and yeast (no colonies on the plate) was considered the MBC and MFC, respectively [22].

### 2.11. Microplate Determination of Plant Extracts’ Antioxydant Activity

2,2-Diphenyl-1-picrylhydrazyl assay

The radical scavenging activity of *J. multifida* leaf extracts against DPPH (2,2-Diphenyl-1-picrylhydrazyl) was assessed using the microplate method outlined by Chokki et al. [23]. For each well, 100 µL of a 50 μM DPPH solution was combined with 100 µL of the plant extracts at a concentration of 200 μg/mL. This combination was then shielded from light and allowed to sit for 20–30 min at room temperature. Absorbance readings were taken at 517 nm using a Tecan Pro microplate reader. For the blank, the sample was substituted with 100 µL of methanol under identical conditions. The effectiveness of *J. multifida* leaf extracts and standard agents (such as ascorbic acid and BHT) in scavenging DPPH radicals was calculated using a previously established formula [24].
$$Inhibitory\;Percentage(\%)=\frac{Blank’s\;absorbance-Sample’s\;absorbance}{Blank’s\;absorbance\;}\times 100$$

The concentration at which 50% inhibition (IC_50_) was achieved was determined using a regression equation derived from the curve that depicted the relationship between the inhibition percentage and the concentration of plant extracts [25].

The 2,2-azinobis (3-ethylbenzthiazoline)-6-sulfonic acid assay

To assess the ABTS radical scavenging potential of *J. multifida* leaf extracts, we followed the methodology outlined by Cudalbeanu et al. [26]. The radical was generated by combining 5 mL of a 7.8 mM ABTS solution with 5 mL of a 140 mM potassium persulfate solution. This mixture was then allowed to rest in the dark at room temperature for 12 h. Subsequently, it was diluted to achieve an absorbance value within the range of 1.1 ± 0.02 at 734 nm. For the test, 100 µL of freshly prepared ABTS solution was combined with 100 µL of *J. multifida* leaf extracts (1000 µg/mL). After allowing the mixture to incubate for 30 min, its absorbance was recorded at 734 nm. Trolox, ranging from 7.8 µg/mL to 62.5 µg/mL, served as the reference for the calibration curve (y=1.3674x+13.732 with R^2^ = 0.998). Outcomes were articulated both in terms of the ABTS radical inhibition percentage and in mole-equivalent Trolox (mol ET/g) according to the formula below. All tests were conducted in triplicate.
$$C=\frac{Co\times fd}{Ci\times M}$$
where: C—concentration of reducing coumpounds in MolET.g^−1^ dry extract; Co—concentration of the sample read; fd—dilution factor of the stock solution; Ci—initial concentration; M—molar mass of Trolox.

Ferric reducing antioxydant power (FRAP)

The procedure was adapted from the method previously detailed by Jones et al. [27], incorporating certain modifications. Specifically, a solution was prepared by combining 100 mL of TPTZ solution (10 mM in 40 mM HCl) with 10 mL of FeCl_3_ (20 mM). From this solution, 200 µL was taken as the working solution and mixed with 50 µL of *J. multifida* leaf extract (1000 µg/mL). This mixture was then incubated at 37 °C for a duration of 10 min. The resultant reaction mixture’s absorbance was subsequently measured at 593 nm. For reference, ascorbic acid was employed as a positive control, with a standard curve ranging from 0 to 250 µg (y=1.3894x+38.444 with R^2^ = 0.99). The extract’s capability to reduce iron (III) to iron (II) was denoted both as a ferric reducing inhibition percentage and in terms of microgram-equivalent ascorbic acid per gram of extract (µg EAA/g sample) according to the formula below. All data points were represented as the average of three separate analyses.
$$C=\frac{Co\times fd}{Ci}$$
where: C—concentration of reducing coumpounds in µgEAA.g^−1^ dry extract; Co—concentration of the sample read; fd—dilution factor of the stock solution; Ci—initial concentration

Lipid peroxidation inhibitory test (LPO)

This activity was assessed using Belem-Kabré et al.’s [18] protocol, with a notable alteration: we substituted rat liver homogenate with egg yolk homogenate. Initially, a solution containing 0.2 mL of *J. multifida* leaf extracts (at 10 mg/mL concentration) or 1.5 mg/mL of ascorbic acid was concocted. To this blend, we introduced 1 mL of a 10% egg yolk homogenate suspended in phosphate-buffered saline (PBS, pH 7.4) and 50 μL of FeCl2 (0.5 mM). This concoction was then left to incubate at 37 °C for an hour. Post-incubation, we sequentially integrated 50 μL of H_2_O_2_ (0.5 mM), 1 mL of trichloroacetic acid (15%), and 1 mL of 2-thiobarbituric acid (0.67%). The mixture was then subjected to centrifugation at 2000 rpm for 10 min, followed by a boiling water bath for 15 min. As a reference point, a control was established using the same method, but replacing the extract with distilled water. Finally, we gauged the absorbance of the mixtures with a spectrophotometer, the Epoch Biotek Instruments model from the USA, targeting a 532 nm wavelength. The extent of lipid peroxidation inhibition was then determined using the following formula:$$\%Inhibition=\frac{Abs\;control-Abs\;sample}{Abs\;control}\times 100$$

### 2.12. Anti-Inflammatory Capacity of J. multifida Extracts

The anti-inflammatory properties of *J. multifida* extracts in vitro were assessed using the methodology outlined by Kabré et al. [28]. In essence, a solution was created by combining 100 µL of *J. multifida* leaf extracts at varying concentrations (ranging from 3.9 to 1000 µg/mL) with 10 µL of egg albumin and 140 µL of phosphate-buffered saline (PBS, pH 6.4). This concoction was then subjected to incubation at 37 °C for a span of 15 min, followed by heating at 70 °C for 5 min. A control setup, or blank, mirrored the same procedure but with the extracts substituted with an equivalent volume of the dilution solvent. Once cooled, the absorbance levels were determined at 660 nm utilizing a microplate reader (Tecan Infinite M 200 Pr, USA). The impact of the *J. multifida* leaf extracts on the thermal denaturation of albumin at 70 °C was denoted by the inhibition rate, which was computed as per the subsequent formula:(1)$$\%\;Inh=\frac{Abs\;C-Abs\;S}{Abs\;C}$$
where: %Inh—inhibition percentage; AbsC—absorbance of control; AbsS—sample absorbance. IC_50_, which denotes the concentration required for 50% inhibition of albumin’s thermal denaturation, was determined from the equation derived from the logarythmic curve (non-linear regression) plotting inhibition percentage against the concentrations of the plant extracts.

### 2.13. Statistical Analysis

Data from the experiments were initially recorded on bench sheets and later input into an Excel 2016 spreadsheet for further processing. GraphPad Prism^®^ version 8.0.2 facilitated both graph creation and statistical analysis. We employed multivariate analysis of variance, succeeded by Tukey’s test, for the statistical evaluations. A *p*-value less than 0.05 was deemed to indicate statistical significance. Data are expressed as the mean ± standard deviation.

## 3. Results

### 3.1. Preliminary Phytochemical Profiling

The initial phytochemical assessment of *J. multifida* powder unveiled the existence of a range of secondary metabolites, as listed in Table 1. Through UV-vis spectroscopy, the presence of polyphenols and flavonoids was verified; they exhibited specific absorption at 340 nm and 280 nm respectively, as illustrated in Figure 1. A qualitative analysis highlighted an unequal distribution of these metabolites across the sample. In fact, 56.25% of the investigated secondary metabolites were found in the *J. multifida* leaf powder. Within the nitrogen compounds category, alkaloids were discernible. The polyphenolic compounds encompassed flavonoids, anthocyanins, leuco-anthocyanins, gallic tannins, and quinonic derivatives. Among the terpenic compounds, only triterpenes were detected. Furthermore, the glycosides category revealed the presence of saponosides and mucilages.

### 3.2. Total Phenolic and Flavonoid, and Condensed and Hydrolyzable Tannin Contents

The contents of the total phenolics and flavonoids, and condensed and hydrolyzable tannins are presented in Table 2.

The concentration of these secondary metabolites varied significantly based on the type of extract (*p* < 0.05). Specifically, the methanolic extract boasted the highest polyphenolic compound content at 45.01 ± 11.87 mgEqGA/g, while the aqueous extract recorded the lowest at 11.25 ± 1.37 mgEqGA/g.

Regarding total flavonoids, the hydro-ethanolic extract exhibited the highest concentration (7.43 ± 0.12 mgEqQ/g). However, this was not statistically different (*p* > 0.05) from the values obtained from both the ethanolic extract (7.18 ± 2.85 mgEqQ/g) and the aqueous extract (6.58 ± 1.36 mgEqQ/g). In this category, the dichloromethane extract yielded the least amount of total flavonoids, registering at 0.32 ± 0.01 mgEqQ/g.

For condensed tannins, the acetone extract led the pack with a concentration of 8.58 ± 0.45 mgEqC/g, followed by the dichloromethane extract (6.79 ± 0.34 mgEqC/g) and the methanolic extract (6.49 ± 0.46 mgEqC/g). As for hydrolysable tannins, they were most abundant in the dichloromethane extract, measuring 4.12 ± 0.29 mgEqGA/g, and least prevalent in the water-ethanolic mix at 2.11 ± 0.12 mgEqGA/g.

### 3.3. Identification of Phenolic Compounds in J. multifida Extract via LC-DAD-MS-ESI^+^ Analysis

Mass spectrometry was employed to discern the polyphenolic compounds within the methanol extract of *J. multifida* leaves. The fragmentation of sample molecules facilitated this identification. Table 3 delineates the identification and quantification of the bioactive compounds extracted from *J. multifida* leaves.

Several phenolic and flavonoid compounds were identified (Figure 2), including 2-Hydroxybenzoic acid, o-Coumaroylquinic acid, Apigenin-apiosyl-glucoside, luteolin-galactoside, luteolin-glucoside, luteolin-rhamnoside, quercetin-glucoside, quercetin-arabinoside, Dicaffeoyquinic acid, and Kaempferol-rhamnoside. Of these, luteolin-rhamnoside was the most prevalent, with a concentration of 19.73 mg/g in *J. multifida* leaf extracts. Following closely was Apigenin-apiosyl-glucoside at 5.58 mg/g. Interestingly, three compounds—luteolin-glucoside (2.17 mg/g), o-Coumaroylquinic acid (2.29 mg/g), and luteolin-galactoside (2.89 mg/g)—were found at nearly equivalent concentrations. Dicaffeoyquinic acid, falling under the Hydroxycinnamic acid subclass, had the lowest detected amount at 0.63 mg/g.

### 3.4. Antimicrobial Activity of J. multifida Leaf Extracts

#### 3.4.1. Susceptibility of Reference Strains to *J. multifida* Leaf Extracts

The antimicrobial activities of aqueous and ethanolic extracts were tested against reference bacterial strains, and the findings are detailed in Table 4. Both extracts exhibited activity against all the tested reference strains, albeit at different concentrations. For the aqueous extract, the Minimum Inhibitory Concentrations (MICs) ranged from 22.67 mg/mL for *S. aureus* to 47.61 mg/mL for *E. coli*, *S. enteritidis*, *L. monocytogenes*, and *C. albicans*. Notably, most of the Minimum Bactericidal Concentrations (MBCs) exceeded 47.61 mg/mL, with *S. aureus* as the exception. In contrast, for the ethanolic extract, the MICs spanned from 5.14 mg/mL up to 34.54 mg/mL for *E. coli*. The MBCs ranged from 22.67 mg/mL (for strains such as *S. aureus*, *S. enteritidis*, *L. monocytogenes*, and *C. albicans*) to 47.61 mg/mL for *E. coli*. When comparing the effects of the two extracts, the aqueous version demonstrated a bacteriostatic effect on the reference strains. Meanwhile, the ethanolic extract exhibited bactericidal properties against four of the reference strains, with *S. aureus* being the exception.

#### 3.4.2. Susceptibility of Meat-Isolated Strains to *J. multifida* Leaf Extracts

Figure 3 shows the appareance of petri dishes for meat-isolated bateria strains.

Inhibition zone diameters varied significantly based on the extract type (*p* < 0.001) and bacterial strain (*p* < 0.001). The aqueous extract did not impact the strains isolated from meat. The dichloromethane extract demonstrated the broadest antimicrobial activity, inhibiting all strains. In contrast, the acetone extract inhibited 71.42% of the strains, the methanol extract inhibited 42.85%, the water-ethanolic extract inhibited 28.57%, and the ethanolic extract inhibited just 14.28%. The dichloromethane extract produced the largest inhibition zone (15.00 ± 0.70 mm) against three strains: *S. aureus*, *S. lentus*, and *S. cohnii*. Impressively, the methanolic extract also exhibited an inhibition diameter of 15.00 ± 1.41 mm against the *S. haemolyticus* strain. Additionally, the dichloromethane extract impeded the growth of the *S. saprophyticus* strain with a diameter of 14.00 ± 1.41 mm, and the *S. simulans* strain with a consistent diameter of 14.00 ± 0.00 mm. Among the foodborne strains, *S. haemolyticus* proved to be the most sensitive to all extracts, while *S. equorum* was the most resilient (as depicted in Figure 4). Specifically, for the highly sensitive *S. haemolyticus* strain, a comparison of the extracts revealed the most pronounced difference in inhibition diameters between the methanolic and ethanolic extracts (*p* < 0.001). Conversely, the smallest difference was observed between the acetone and methanol extracts (*p* = 0.01).

Similar to the inhibition diameters, the MICs of the active extracts varied depending on the bacterial strains and extract types. The methanolic extract showed the lowest MIC at 10 mg/mL for the *S. cohnii* strain, and its MBC was also the lowest at 20 mg/mL. This suggests that the methanolic extract exhibits a bactericidal effect on the *S. cohnii* strain, as presented in Table 5. For the acetone extract, it had an MIC of 10 mg/mL, yet its MBC exceeded 20 mg/mL for all susceptible strains. In the case of the dichloromethane extract, MIC values ranged from 2.5 mg/mL to 5 mg/mL, and MBC values varied between 5 mg/mL and 20 mg/mL. Notably, the dichloromethane extract was the most potent against foodborne strains, demonstrating a bactericidal action against *S. equorum*, *S. saprophyticus*, *S. haemolyticus*, and *S. lentus*. Although it showed a bacteriostatic effect on the *S. simulans* strain, its MBC was recorded at 10 mg/mL. On the other hand, the ethanolic extract had a bactericidal impact on the *S. haemolyticus* strain. However, the water-ethanolic extract did not exhibit any bactericidal activity against the strains that were sensitive to it.

### 3.5. Antioxydant Activity of J. multifida Leaf Extracts

The antioxidant capacity of the extracts was assessed using DPPH radical, ABTS radical cation, ferric ion (FRAP), and lipid peroxidation (LPO) reduction methods. The results are illustrated in Figure 5. Among the *J. multifida* extracts, the ethanolic extract exhibited the most potent DPPH radical scavenging activity, as indicated by its lowest IC_50_ value (0.72 ± 0.03 mg/mL). The methanolic extract followed closely, registering an IC_50_ of 0.87 ± 0.01 mg/mL. On the other hand, the dichloromethane extract demonstrated the least antioxidant activity using the DPPH method, with a higher IC_50_ value of 2.96 ± 0.37 mg/mL. The antioxidant capacities of the water-ethanolic (IC_50_ = 1.01 ± 0.01 mg/mL) and acetone extracts (IC_50_ = 1.85 ± 0.08 mg/mL) were found to be intermediate. For comparison, the reference molecules, ascorbic acid and BHT, showcased robust DPPH radical reduction capabilities, recording IC_50_ values of 0.02 ± 0.00 mg/mL and 0.09 ± 0.01 mg/mL, respectively (see Figure 5a). Moreover, the methanolic extract had the highest ferric ion inhibition percentage, standing at 87.28 ± 1.01%, and a concentration of 46.23 ± 1.10 µgEAA/g (as shown in Figure 5b). This was followed by the hydroethanolic extract, which had an inhibition percentage of 85.05 ± 0.04% and a concentration of 33.55 ± 0.03 µgEAA/g. Conversely, the dichloromethane extract had the least ferric ion reduction capability (63.79 ± 0.54%), mirroring its performance in the DPPH reduction.

When examining the ABTS radical cation inhibition capacity (refer to Figure 5c), the activity (expressed as inhibition percentage) of the extracts decreased in the following sequence: acetone (45.49 ± 2.10%) < ethanol (46.58 ± 11.59%) < dichloromethane (47.15 ± 2.02%) < methanol (63.78 ± 1.16%) < water-ethanol (84.26 ± 1.68%). Notably, when using the ABTS method, the water-ethanolic extract demonstrated a higher reducing power (0.49 ± 0.11 mol ET/g) compared to the methanolic extract (0.25 ± 0.03 mol ET/g), which is a deviation from the results seen with the FRAP method. In the lipid peroxidation assay, the methanolic extract displayed the highest inhibition rate (42.19 ± 2.60%), closely followed by the water-ethanolic extract (40.00 ± 9.07%). There was no significant difference (*p* > 0.05) between these two extracts and the potency of ascorbic acid (used as a reference), which exhibited an inhibition rate of 48.59 ± 4.21%. Among all, the acetone extract showed the lowest inhibition rate at 28.80 ± 2.89%. When compared to certain extracts, ascorbic acid significantly outperformed the acetone extract (*p* = 0.0019), dichloromethane (*p* = 0.0024), and ethanol extract (*p* = 0.0026) in lipid peroxidation inhibition.

### 3.6. Antiinflammatory Activity of *J. multifida* Leaf Extracts

The ability of *J. multifida* leaf extracts to prevent thermal protein denaturation (specifically albumin) was assessed at 660 nm. A dose-dependent response was evident in the *J. multifida* extracts tested, as depicted in Figure 6. The water-ethanolic extract displayed the highest inhibition percentage against albumin denaturation, reaching 97.31 ± 0.35% at a concentration of 1000 µg/mL. Following closely was the methanolic extract at the same concentration, which inhibited denaturation by 95.35 ± 1.05%. In contrast, the dichloromethane extract presented the least effective inhibition at 80.64 ± 1.32%. 

Concerning IC_50_ values, the same trend was observed. Thus, the water-ethanolic extract showed the lowest IC_50_ (17.37 ± 0.36 µg/mL), suggesting a stronger anti-inflammatory activity. Furthermore, no difference (*p* > 0.05, as indicated by ANOVA) was observed between ethanolic and methanolic extracts. Both the dichloromethane and acetone extracts registered the highest IC_50_ values, at 263.77 ± 5.59 µg/mL and 221.41 ± 3.13 µg/mL, respectively, indicating lower potency.

## 4. Discussion

Phytochemical analysis reveals that the *J. multifida* leaf powder is a rich source of secondary metabolites. The presence of these metabolites suggests potential biological activities in the tested extracts. Initial screenings identified the presence of alkaloids within the nitrogen compounds group. This is consistent with findings of Rampadarath et al. [29] and Hanafi et al. [30], who observed similar results for *J. multifida* leaves sourced from Mauritius and Indonesia, respectively. Alkaloids are recognized for their broad spectrum of biological activities, including antiviral, antibacterial, anti-inflammatory, antioxidant, and anticancer properties [31]. Our study also detected the presence of tannins, flavonoids, and saponins in the leaf powder. Aiyelaagbe [13] in Nigeria, Nwokocha et al. [32] in Niger, and Chokchaisiri et al. [33] in Thailand have previously reported the presence of flavonoids and saponins in the same plant species. Both tannins and flavonoids are acknowledged for their antibacterial, antiviral, antifungal, and antioxidant properties. Notably, these compounds have been found to foster tissue regeneration in cases of superficial burns [34]. Furthermore, terpenic compounds were identified in *J. multifida* leaves. Various studies have documented the occurrence of monoterpenes, diterpenes, and triterpenes in the plant [35,36]. Like many natural compounds, terpenes are renowned for their antimicrobial activities, especially against antibiotic-resistant bacteria. They have the capability to cause cell rupture and inhibit protein and DNA synthesis [37,38].

Zhao et al. [39] highlighted the significant role terpene compounds play in promoting human health. These compounds are renowned for treating various human diseases, boasting properties such as antimicrobial, anticancer, anti-inflammatory, antioxidant, and neuroprotective properties. In the *J. multifida* leaf powder, the absence of cyanogenic derivatives is crucial as these can induce toxicity through the release of cyanide ions, leading to symptoms such as respiratory acceleration, depression, dizziness, headaches, disturbances in consciousness, and even coma [34]. The quantification of total phenolics, flavonoids, and both condensed and hydrolyzable tannins corroborate the results from qualitative tests.

A mass spectrometry technique using electrospraying in positive ionization mode was deployed to pinpoint polyphenolic compounds in the methanolic *J. multifida* leaf extract. This led to the identification of various phenolic and flavonoid compounds, including 2-Hydroxybenzoic acid, o-Coumaroylquinic acid, Apigenin-apiosyl-glucoside, luteolin-galactoside, and others. Our research is among the few studies employing the LC-MS-ESI technique to analyze *J. multifida* compounds, as opposed to the methods used for species such as *J. curcas* and *J. gossipifolia*. However, recent studies [30] have utilized the GC-MS technique to examine the volatile compounds of *J. multifida*, revealing the rich diversity of secondary metabolites that contribute to its biological activities.

Indeed, the tested extracts displayed potent antimicrobial properties. For instance, the ethanolic extract exhibited bactericidal effects against reference strains such as *E. coli* and *S. enteritidis*. The dichloromethane extract also showed bactericidal properties against meat-isolated strains such as *S. equorum* and *S. haemolyticus*. Other studies from Indonesia and Togo [9,29] have similarly reported the antimicrobial prowess of *J. multifida* extracts.

This antimicrobial activity might be attributed to the tannins present, which exhibit antibacterial properties against both Gram-positive and Gram-negative bacteria. Literature suggests [40] that tannins counteract bacterial growth via multiple mechanisms, including enzyme complexation and bacterial substrate interaction, and by altering cell membrane permeability. Additionally, the antimicrobial potency of these extracts might also be credited to the alkaloids and Dicaffeoylquinic acid discovered in the *J. multifida* leaves. Considering bacterial cell membranes comprise lipid bilayers and glycoproteins, some alkaloids reportedly embed into the lipid bilayers and interact with the sugars present, causing chemical rearrangements that lead to cell lysis [41]. Dicaffeoylquinic acids, found in plants such as *Youngia japonica*, have also been acknowledged for their antibacterial activities [42,43,44].

Beyond antimicrobial effects, *J. multifida* leaf extracts have demonstrated antioxidant and anti-inflammatory properties. Each extract tested exhibited notable antioxidant activity, evidenced by the reduction in DPPH and ABTS radicals, ferrous ion (FRAP), and lipid peroxidation (LPO). This activity can be attributed to the compounds identified using the LC-MS-ESI technique in this study. Notably, luteolin-rhamnoside was the most abundant compound in the *J. multifida* leaf extracts, with a concentration of 19.73 mg/g. Other compounds, including luteolin-glucoside, quercetin-glucoside, quercetin-arabinoside, and Kaempferol-rhamnoside, were also detected.

The flavone group, which encompasses luteolin and luteolin-glucoside, and the flavonol group, which includes quercetin and Kaempferol, are recognized for their diverse beneficial properties. This is due to their ability to undergo structural modifications through chemical reactions such as hydroxylation, O-/C-glycosylation, O-methylation, and acylation [45]. These compounds are widely acknowledged as potent anti-inflammatory and antioxidant agents [46,47]. Considering the soluble messengers facilitating communication between immune cells, compounds such as luteolin and luteolin-glucoside in *J. multifida* extracts have been highlighted by various researchers [47,48] for their capacity to regulate the production of proinflammatory cytokines such as TNF-α, IL-1β, and IL-6. This regulatory function directly impacts inflammatory pathways [47,48], endowing luteolin and luteolin-glucoside with their anti-inflammatory attributes.

Although our study did not delve into this mechanism, *J. multifida* extracts showed commendable anti-inflammatory activities, evident by the inhibition of protein denaturation (albumin) with an IC_50_ value of 17.37 ± 0.36 µg/mL. Other research on experimental animal models and HEKn cells, as documented by Palombo et al. [49] and Caporali et al. [50], also underscored the anti-inflammatory potential of luteolin-glucoside, a compound present in *J. multifida* leaf extracts. These findings collectively validate the anti-inflammatory prowess of *J. multifida* extracts.

J. multifida leaf extracts exhibited robust antioxidant activity, as demonstrated through four distinct methods: DPPH, FRAP, ABTS, and LPO. This potent antioxidant capacity is likely attributed to the presence of compounds such as Kaempferol, luteolin, and quercetin, all renowned for their exceptional antioxidant properties.

Of the compounds identified in *J. multifida* leaf extracts, luteolin’s antioxidant properties have been extensively studied. Research involving in vitro and in vivo models using NRK-52E rat kidney cells treated with ochratoxin A (OTA) found that luteolin rejuvenated the antioxidant capacity of kidney cells by activating Nrf2 [51]. Further, Alekhya et al. [52] highlighted luteolin’s dual role in diminishing lipid peroxidation associated with the generation of pro-inflammatory lipids and mitigating DNA damage. This is achieved by upregulating both HO-1 (heme oxygenase) and Nrf2, vital for countering oxidative stress.

Additionally, compounds such as luteolin-glucoside, derivatives of Kaempferol, and quercetin, along with its glucosides, have been documented for their capabilities to neutralize the deleterious effects of free radicals that induce oxidative stress. This oxidative imbalance is a known precursor to various diseases across different cellular systems [53,54,55].

## 5. Conclusions

*J. multifida* leaves are rich repositories of secondary metabolites, showcasing a range of bioactive compounds from alkaloids to terpenes. These compounds are renowned for their multifaceted biological roles, notably in antimicrobial, antioxidant, and anti-inflammatory capacities.

The extracts from *J. multifida* leaves have been heralded for their notable antioxidant prowess, largely owing to the compounds such as luteolin, Kaempferol, and quercetin. Luteolin, for instance, is acclaimed for its potential in revitalizing antioxidant activities in kidney cells, serving as a defense against oxidative stress. Additionally, molecules such as luteolin and its glucoside derivative stand out for their anti-inflammatory contributions. They modulate the production of specific pro-inflammatory cytokines such as TNF-α, IL-1β, and IL-6, indicating their direct impact on inflammatory pathways. Their proficiency in inhibiting protein denaturation further emphasizes their potential.

In terms of antimicrobial capabilities, *J. multifida* leaf extracts display a commendable performance, particularly against certain benchmark bacterial strains. The antimicrobial prowess is primarily attributed to the tannins present in the extracts. Other compounds, including alkaloids and Dicaffeoylquinic acid, amplify this antimicrobial efficacy. The plethora of these dynamic bioactive compounds accentuates the therapeutic promise of *J. multifida* extracts. Their impressive antioxidant, anti-inflammatory, and antimicrobial traits present a compelling case for their integration into medicinal or nutraceutical formulations.

While sophisticated techniques such as LC-MS-ESI have delved into analyzing *J. multifida* compounds, they have not been as extensively used as those of its relatives, *J. curcas* and *J. gossipifolia*. This hints at a vast realm of untapped research potential concerning *J. multifida*, poised to uncover its full therapeutic horizon.

In summary, a comprehensive assessment of *J. multifida* leaf extracts reveals their stature as powerhouses of potent bioactive entities. With its array of health benefits, *J. multifida* stands out as a promising subject for future research and potential health-oriented innovations.

## Figures and Tables

**Figure 1 life-13-01898-f001:**
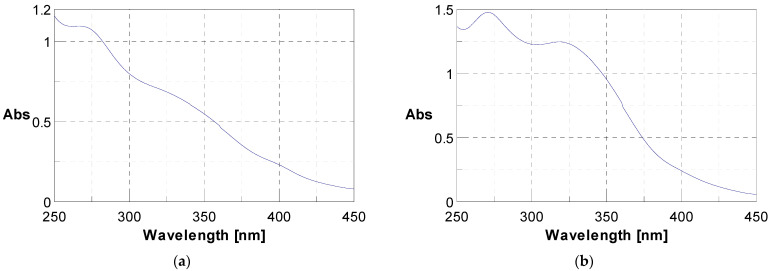
UV-vis spectra showing the presence of polyphenols at 280 nm (**a**) and flavonoids at 340 nm (**b**).

**Figure 2 life-13-01898-f002:**
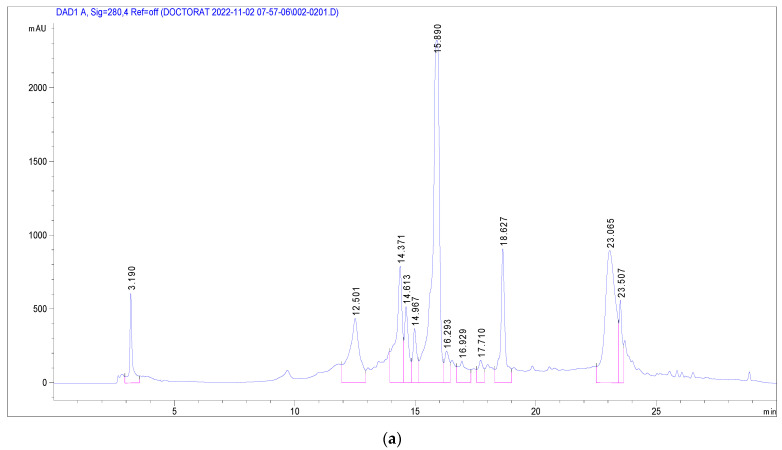
Chromatographic profile at 280 nm (**a**) and 340 nm (**b**) of *J. multifida* methanolic extract in electrospray positive ionization mode.

**Figure 3 life-13-01898-f003:**
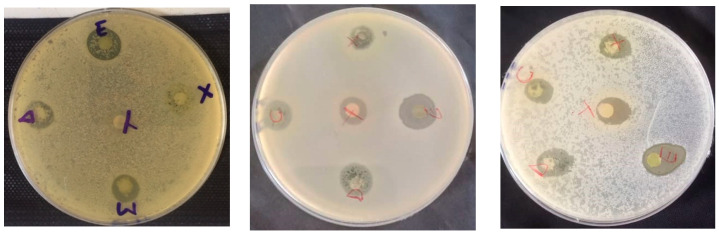
Appearance of some petri dishes.

**Figure 4 life-13-01898-f004:**
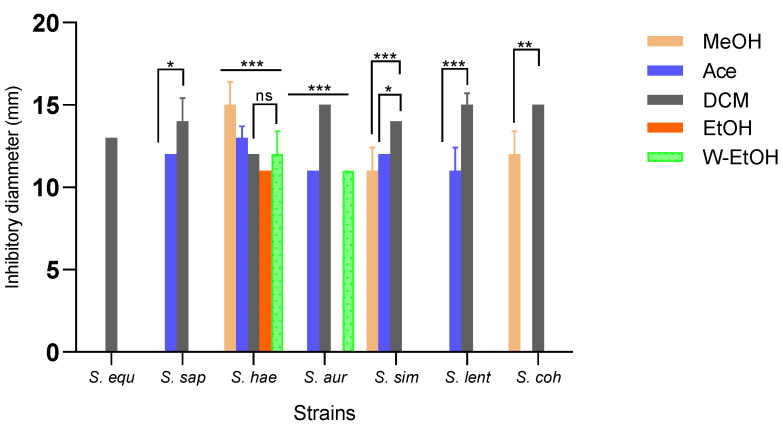
Medium inhibitory diameter zone of *J. multifida* leaf extracts on meat-isolated *Staphylococcus* strains. MeOH: Methanol, Ace: Acetone; DCM: Dichlorometane; EtOH: Ethanol; W-EtOH: Water-Ethanol; *S. equ*: *Staphylococcus equorum*; *S. sap*: *Staphylococcus saprophyticus*; *S. hae*: *Staphylococcus haemolyticus*; *S. aur*: *Staphylococcus aureus*; *S. sim: Staphylococcus simulans*; *S. len*: *Staphylococcus lentus*; *S. coh*: *Staphylococcus cohnii*. *: *p* < 0.05; **: *p* < 0.01; ***: *p* < 0.001; ns: *p* > 0.05.

**Figure 5 life-13-01898-f005:**
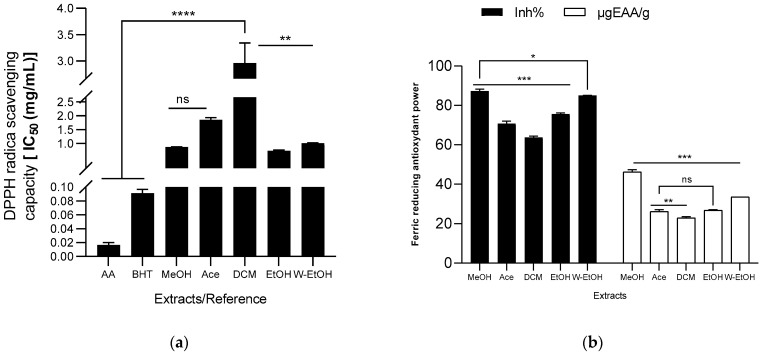
Antioxydant activity of *J. multifida* extract using DPPH (**a**), FRAP (**b**), ABTS (**c**), and LPO (**d**) methods. EAA: equivalent ascorbic acid, ET: equivalent trolox, MeOH: Methanol, Ace: Acetone; DCM: Dichlorometane; EtOH: Ethanol; W-EtOH: Water-Ethanol; *: *p* < 0.05; **: *p* < 0.01; ***: *p* < 0.001; ****: *p* = 0.0001; ns: *p* > 0.05.

**Figure 6 life-13-01898-f006:**
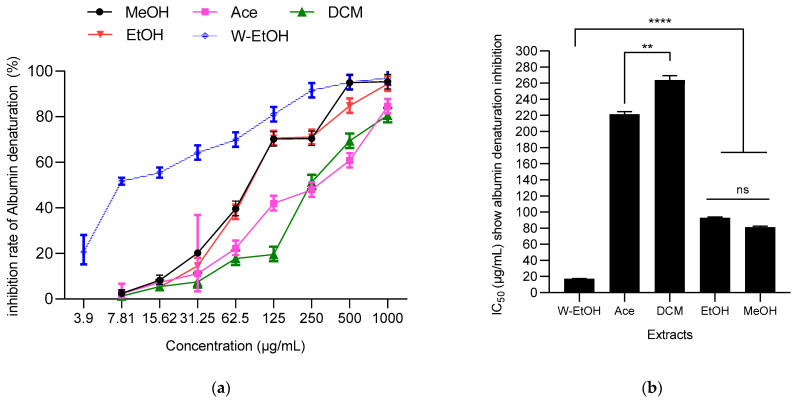
J. multifida extracts’ inhibition rate of albumin denaturation (**a**) and their IC_50_ (**b**). MeOH: Methanol, Ace: Acetone; DCM: Dichlorometane; EtOH: Ethanol; W-EtOH: Water-Ethanol; **: *p* < 0.01; ****: *p* = 0.0001; ns: *p* > 0.05.

**Table 1 life-13-01898-t001:** Phytochemical constituents of *J. multifida* leaf powdered samples.

Group of Compounds	Class	*J. multifida*
Nitrogenous compound	Alkaloids	+
Polyphenolics compound	Tanin catéchique	−
	Tanin gallique	+
	Flavonoids	+
	Anthocyans	+
	Leuco-anthocyane	+
	Coumarin	−
	Quinonics derivate	+
Terpenic compound	Triterpenoids	+
	Steroids	−
	Cardenolids	−
Heterosides	Saponosids (IM)	+ (112)
	Cyanogenics derivate	−
	Reducing compounds	−
	Free anthracénics	−
	Mucilags	+

(+): Presence of secondary metabolite. (−): Absence of secondary metabolite. (IM): Index mouss.

**Table 2 life-13-01898-t002:** Phenolic, flavonoid, and condensed and hydrolyzable tannin contents of *J. multifida* leaf extracts.

Solvents	Polyphenols (mgEqGA/g)	Flavonoids (mgEqQ/g)	Condensed Tannins (mgEqC/g)	Hydrolysable Tannins (mgEqGA/g)
Water	11.25 ± 1.37	6.58 ± 1.36	2.05 ± 0.28	2.18 ± 0.15
Ethanol	23.03 ± 6.9	7.18 ± 2.85	5.49 ± 0.35	3.03 ± 0.38
Water-Ethanol	21.57 ± 2.20	7.43 ± 0.12	4.78 ± 0.42	2.11 ± 0.12
Methanol	45.01 ± 11.87	3.65 ± 0.09	6.49 ± 0.46	3.31 ± 0.52
Acetone	26.83 ± 0.27	0.90 ± 0.03	8.58 ± 0.45	3.19 ± 0.11
Dichloromethane	12.86 ± 0.44	0.32 ± 0.01	6.79 ± 0.34	4.12 ± 0.29

**Table 3 life-13-01898-t003:** Phenolic compounds from *J. multifida* leaf extracts identified via LC-DAD-MS-ESI^+^ analysis.

Peak No.	R_t_ (min)	UV λ_max_ (nm)	[M + H]^+^ (*m*/*z*)	Phenolic Compound	Subclass	Content (mg/g)
1	3.19	270	139	2-Hydroxybenzoic acid	Hydroxybenzoic acid	1.250
2	12.50	333	339	*o*-Coumaroylquinic acid	Hydroxycinnamic acid	2.291
3	14.37	330, 280	565, 271	Apigenin-apiosyl-glucoside	Flavone	5.587
4	14.61	350, 260	449, 287	Luteolin-galactoside	Flavone	2.890
5	14.96	350, 260	449, 287	Luteolin-glucoside	Flavone	2.173
6	15.88	350, 260	433, 287	Luteolin-rhamnoside	Flavone	19.732
7	16.31	360, 260	465, 303	Quercetin-glucoside	Flavonol	1.092
8	16.52	360, 260	434, 303	Quercetin-arabinoside	Flavonol	0.779
9	16.93	332	517	Dicaffeoyquinic acid	Hydroxycinnamic acid	0.636
10	17.71	350, 250	433, 287	Kaempferol-rhamnoside	Flavonol	1.223
				Total phenolics		37.652

**Table 4 life-13-01898-t004:** Minimum Inhibitory Concentration (MIC) and Bactericidal/Fungicidal Concentration (MBC/MFC) of *J. multifida* extracts.

Extracts	Parameters (mg/mL)	*Staphylococcus aureus* ATCC 6538P	*Escherichia coli* ATCC 25922	*Salmonella enteritidis* ATCC 13076	*Listeria monocytogenes* ATCC 19114.	*Candida albicans* ATCC 10231
Water	MIC	22.67	47.61	47.61	47.61	47.61
	MBC	47.61	>47.61	>47.61	>47.61	>47.61
	MBC/MIC	2.10	nd	Nd	nd	nd
Ethanol	MIC	5.14	34.54	22.67	22.67	22.67
	MBC	22.67	47.61	22.67	22.67	22.67
	MBC/MIC	4.41	1.37 *	1 *	1 *	1 *

The ratio MBC/MIC value with * = Bactericidal effects and the ratio MBC/MIC value without * = Bacteriostatical effects.

**Table 5 life-13-01898-t005:** Minimum Inhibitory Concentration (MIC) and Minimum Bactericidal Concentration (MBC) of *J. multifida* extracts on meat-isolated *Staphylococcus*.

Extracts	Parameters (mg/mL)	*S. equ*	*S. sap*	*S. hae*	*S. aur*	*S. sim*	*S. len*	*S. coh*
MeOH	MIC	-	-	20	-	20	-	10
	MBC	-	-	>20	-	>20	-	20
	MBC/MIC	-	-	-	-	-	-	2 *
Ace	MIC	-	10	20	20	20	20	-
	MBC	-	>20	>20	>20	>20	>20	-
	MBC/MIC	-	-	-	-	-	-	-
DCM	MIC	2.5	2.5	5	2.5	2.5	2.5	5
	MBC	5	5	5	20	10	5	20
	MBC/MIC	2 *	2 *	1 *	8	4	2 *	4
EtOH	MIC	-	-	10	-	-	-	-
	MBC	-	-	20	-	-	-	-
	MBC/MIC	-	-	2*	-	-	-	-
W-EtOH	MIC	-	-	20	20	-	-	-
	MBC	-	-	>20	>20	-	-	-
	MBC/MIC	-	-	-	-	-	-	-

MeOH: Methanol, Ace: Acetone; DCM: Dichlorometane; EtOH: Ethanol; W-EtOH: Water-Ethanol; *S. equ*: *Staphylococcus equorum*; *S. sap*: *Staphylococcus saprophyticus*; *S. hae*: *Staphylococcus haemolyticus*; *S. aur*: *Staphylococcus aureus*; *S. sim*: *Staphylococcus simulans*; *S. len*: *Staphylococcus lentus*; *S. coh*: *Staphylococcus cohnii*. The ratio MBC/MIC value with * = Bactericidal effects and without * = Bacteriostatical effects.

## Data Availability

Not applicable.

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
