# Peer review of "Polyphenol Analysis via LC-MS-ESI and Potent Antioxidant, Anti-Inflammatory, and Antimicrobial Activities of Jatropha multifida L. Extracts Used in Benin Pharmacopoeia"

_life, 2023, doi:10.3390/life13091898_

Round 1

Reviewer 1 Report

The manuscript “Polyphenols analysis by LC-MS-ESI and potent antioxidant, anti-inflammatory, antimicrobial activities of Jatropha mul-tifida L. extracts used in Benin pharmacopoeia” is mostly original and actual study written in a clear form. The work considers chemical composition of Jatropha multifida and its applicability for therapeutics. This is not a pioneer study, but it definitely brings a significant contribution in field of green pharmacy. The conclusions are useful to pharmacy and medicine specialists. The main importance of the article is qualitative and quantitative phytochemical analysis of Jatropha multifida.

The study does not require any experimental revision, but the manuscript must be improved for the better clarity:

1. Compounds obtained from Sigma-Aldrich (2.1) must be mentioned with corresponding Cat. Nos. since Sigma-Aldrich provides compounds of various purity, purpose and history of preparation.

2. In Table 3, it must be clarified what do second numbers after comma mean and what is the difference between italics and regular text for these numbers.

3. In table 4, the same symbol (*) was used for different comments.

4. Dichloromethane’s abbreviation is not “dich”, the common and widely used is “DCM”. The same is about acetone: “Ace” but not “Act”. It must be replaced everywhere in text.

5. The text contains some grammatical mistakes: “Index mouss”, “oxydatif”. It must be improved.

Reviewer 2 Report

Dear Authors,

Data in your ms have the particular value of presenting the analytical characterization of phytochemicals present in leaves of J. multifida employing HPLC combined with mass spectrometry. Locals use this specie to treat several affections, including wounds, skin and gastrointestinal infections. At this point, I recommend elevating this point by giving some historical data about the use of this medical plant. Is it belong to an ancestry medical use? Is there an anthropologic record of its use? Is it possible to put a paleontological glance at the relation of ancestors with this plant? 

Following the point that the plant was used for infections, the antimicrobial effect of five extracts on a panel of bacteria, which is, among the other activities presented, the only one that shows interactions of the extracts with living organisms; all the others are in principle very chemicals (DPPH, FRAP) or physicochemical (albumin denaturation). Thus, I would put data in the following order: 1) Extract characterization as was presented, followed by antioxidant properties of these extracts. 2) bactericidal/fungicide activity 3) albumin denaturation assay.

In a more technical dimension, I noted some missing information that should be well-explained in material and methods. For instance, EC50 estimation from doses-response curves requires the data fitting to a function; which function did you choose to obtain the EC50 you reported? That should be explicit on the legend figure or in materials and methods. As indicated above, I would like to see, and probably the people who will read your ms, the photographs of Petri plates exposed to filters with extracts that demonstrate the bactericidal/fungicide activity of the extracts. Remember to include the control filters with the solvents.

Regarding anti-inflammatory activity, the albumin denaturation assay is not convincing. Although the effect of extracts on protein denaturation is evident, the connection with an immune cells response is quite far. On the other hand, luteolin-glucoside has the potential to be responsible for the anti-inflammatory effect. Still, it is more concentrated in the extract with more anti-inflammatory activity (fig 5B). In this figure, some EC50 values, the lowest values, do not correspond to the dose-response curve exhibited (Fig 5A). Check that.     

Finally, the text within the ms has many typographical, spelling and other errors that must be corrected. 

English language must be refined. Many errors are in the text; professional editing is highly recommended.  

Reviewer 3 Report

Please find the attachment

Extensive editing of English language required

Reviewer 4 Report

In the proposed paper: “Polyphenols analysis by LC-MS-ESI and potent antioxidant, anti-inflammatory, antimicrobial activities of Jatropha multifida L. extracts used in Benin pharmacopoeia” the biological properties of the plant Jatropha multifidi are presented. The authors indicated several properties of the plant extracts such as antioxidant, anti-inflammatory and antimicrobial properties. In addition, some of the phenolic and flavonoid compounds were identified and the total content of some of the active compounds was determined. The work presented could be a valuable contribution to the field, however, there are some issues that should be addressed before the work can be accepted.

Some remarks:

English should be corrected throughout the whole text.

Please ensure that the spelling of the bacteria names is correct throughout the entire text.

Please ensure that all abbreviations are explained when they appear in the text for the first time.

Materials and Methods section: Please specify the concentration of the plant extract used in all mentioned experiments. Section 2.4 shows that the extracts were dried; however, in the experiments, specified volumes were used.

Section 3.3: The authors first mention that the identification was done by the fragmentation of the molecules, and in the next sentence, they say that the chromatograms were recorded with the DAD detector, which has no logical sense.

Figure 2: It is not correct to present chromatogram from DAD detector as the MS chromatogram. Please either change the caption of the figure or the chromatogram to one from MS detector.

Analytical standards used for LC-DAD-MS are not listed in the Material and Methods section. If the standards were not used, please explain how the quantitation was performed.

Please consider changing the abbreviation of dichloromethane to DCM.

Both grammar and spelling should be improved.

Round 2

Reviewer 2 Report

Dear authors, 

The new version of the ms is improved. However, I still need an explanation for the EC50 values shown for MeOH (fig 6B). If raw data from dose-response curves presented in the fig 6A were used, it is evident that the concentration range did not cover the necessary range to do a reasonable estimation of the EC50, curves started with concentration from 31.25 ug/ml. This issue is particularly relevant for the EC50 with MeOH. 

Observing the dose-response curve in fig 6A shows that data do not follow a line. The curves tend to saturate at higher concentrations, as anybody would expect; why was linear regression used to estimate EC50? That is incorrect; there are other math functions to better fit this type of data, but never a linear regression—a rectangular hyperbole, to mention the simplest one.

Experimental data missing combined with conceptual errors are not welcome, especially when EC50 values are presented.  

Author Response

See attached and in text

Reviewer 3 Report

accept for puplication

please carefully review for minor typos 

Author Response

We would like to thank reviewer for this positive comment. The minor typos have been correctef